# PHASEFUSION:
# A DIFFUSION-BASED PERIODIC PARAMETERIZED
# MOTION GENERATION FRAMEWORK

## ABSTRACT

In this study, we introduce a learning-based method for generating high-quality
human motion sequences from text descriptions (e.g., "A person walks forward").
Existing techniques struggle with motion diversity and smooth transitions in gen-
erating arbitrary-length motion sequences, due to limited text-to-motion datasets
and the pose representations used that often lack expressiveness or compactness.
To address these issues, we propose the first method for text-conditioned human
motion generation in the frequency domain of motions. We develop a network
encoder that converts the motion space into a compact yet expressive parameterized
phase space with high-frequency details encoded, capturing the local periodicity
of motions in time and space with high accuracy. We also introduce a conditional
diffusion model for predicting periodic motion parameters based on text descrip-
tions and a start pose, efficiently achieving smooth transitions between motion
sequences associated with different text descriptions. Experiments demonstrate
that our approach outperforms current methods in generating a broader variety of
high-quality motions, and synthesizing long sequences with natural transitions.

## 1 INTRODUCTION

Virtual human characters play a crucial role in various computer graphics and robotics applications,
such as AR/VR, gaming, and simulation of human-robot interactions. Generating natural human
motions with conventional computer graphics techniques involves a tedious and time-consuming
process, requiring complex motion capture recordings, along with manual editing and synthesis of
motions based on selected key frames by animation experts. To address these challenges and make
the process more accessible to non-experts, recent studies (Tevet et al., 2022; Zhang et al., 2022;
Dabral et al., 2023) have introduced text-conditioned deep generative models to generate human
motions based on text descriptions (e.g., keywords of actions like "walk" and "jump", and short
sentences like "A person rotates 360 degrees and walks forward"). While there has been significant
progress, current methods still face challenges in generating diverse animations, ensuring smooth
transitions between motion sequences associated with different text descriptions, and creating natural
looping animations. These challenges primarily stem from limited paired language-motion training
data and the pose representations used which are often either lacking in expressiveness or inefficient
in compactness for generating motion transitions through blending.

In a distinct area of character animation generation, which is under the user's specific locomotion
control (e.g., trajectory direction), some studies (Holden et al., 2017; Zhang et al., 2018; Starke
et al., 2022) explore the idea of identifying smooth transitions in a transformed periodic phase space.
However, these methods are only trained on a specific type of motions (e.g., walking) and their
performance degrades significantly when training on diverse motions. Additionally, these methods
often fail to capture high-frequency information in the synthesized motions. Furthermore, applying
these methods to human motion generation with semantic text control is challenging due to the
ambiguous mapping from text descriptions to the periodic phases.

In this work, we propose the first method for text-conditioned human motion generation in the
frequency domain. Similar to DeepPhase Starke et al. (2022), we presume that the full-body
movements of interest inherently exhibit local periodicity in both time and space, and we learn a

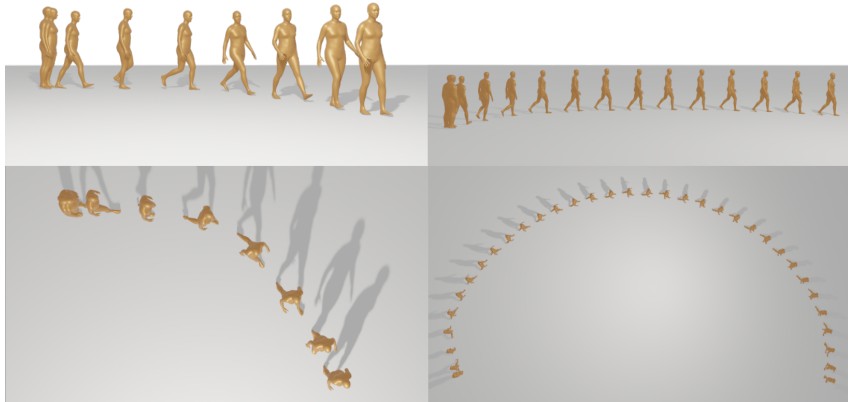

Figure 1: **Left:** Results generated from "the person is walking forward turning right." in default length. **Right:** Extended results using phase repetition. The character can continuously walk to the right without the need to halt and restart.

periodic parameterized phase space with an autoencoder to encode the local periodicity of each body part over time in motions. To tackle the aforementioned issues in DeepPhase Starke et al. (2022), we introduce a designated frequency set that includes high frequencies to enhance the encoding capability for motion details. By encoding motions in this periodic phase space, we alleviate the need for a large amount of training data. Next, to handle the ambiguity in text-to-motion mapping, we introduce a conditional diffusion model that predicts periodic phase parameters to generate motions conditioned on text descriptions and the start pose. Unlike previous approaches (Starke et al., 2022; Holden et al., 2017), we eliminate the requirement for additional network modules to align the phases between two poses for smooth transitions. Instead, we can produce natural motion transitions efficiently by conditioning on the start pose. We have evaluated our approach using two large human motion datasets with text annotations. The results demonstrate the superiority of our approach over existing methods in terms of the quality and diversity of the generated motions, as well as in facilitating smooth and natural motion transitions.

In summary, our contributions are:

1. We propose the first method for text-conditioned human motion generation in the frequency domain, which enables efficient synthesis of diverse and arbitrary-length human pose sequences with smooth and natural transitions.

2. We introduce a novel motion encoding method that maps human motions into a compact yet expressive periodic parameterized space with high-frequency details encoded, thereby enhancing the capability to learn high-quality human motions from diverse motion datasets.

3. We design a new conditional motion diffusion model conditioned on a start pose for parameterized phase prediction, effectively achieving seamless transitions between motion sequences associated with different text descriptions.

## 2 RELATED WORK

### 2.1 MOTION GENERATIVE MODELS

The stochastic nature of human motion has driven the application of generative models in motion generation tasks. Prior works have shown impressive results in leveraging generative adversarial networks (GANs) to sequentially generate human motion Barsoum et al. (2017); Cai et al. (2018). Variational autoencoder (VAE) and transformer architectures have also been widely adapted for their capability of improving the diversity of generated motion sequences Ikhsanul Habibie & Komura (2017); Guo et al. (2020; 2022); Petrovich et al. (2022); Athanasiou et al. (2022). Most recently,Zhang et al. (2023a) proposed T2M-GPT, a framework that learns a mapping between human motion and discrete codes in order to decode motion from token sequences generated by a Generative Pre-trained Transformer. Large Language Models have also been leveraged for text-to-motion tasks to boost

zero-shot capabilities Zhang et al. (2023c); Jiang et al. (2023); Kalakonda et al. (2023). Diffusion models, with their strong ability to model complex distributions and generate high-fidelity results, have become increasingly popular for motion generation, especially for context-conditioned schemes Yuan et al. (2022); Zhang et al. (2022); Dabral et al. (2023); Kim et al. (2023). With a transformer-encoder backbone, Tevet et al. (2022) presented a light-weight diffusion model that predicts samples instead of noise for efficient text-to-motion generation. Zhang et al. (2023b) enhanced the quality of generation for uncommon conditioning signals by integrating a retrieval mechanism into a diffusion-based framework for refined denoising. Although diffusion-based approaches for motion generation have achieved outstanding results, the synthesis of long, arbitrary-length sequences remains limited in quality and computationally expensive. Recently, Chen et al. (2023) greatly improved the efficiency of the diffusion process by introducing a motion latent space, providing an avenue for addressing computational challenges. However, similar to prior approaches, limited data still restricts the capability of generating naturally aligned movements for arbitrary time sequences.

## 2.2 PHASE IN MOTION SYNTHESIS AND CONTROL

Previous works in the field of computer graphics have developed approaches using periodic signals to enhance movement alignment over time, incorporating phase variables that represent the progression of motion. Phase-functioned Neural Networks Holden et al. (2017), a notable instance of such approaches, adjust network weights based on a phase variable, defined by interactions between the foot and the ground during leg movement. This principle has been expanded to include additional types of complex movements Zhang et al. (2018) Starke et al. (2019), and for complex movements involving multiple contact points Starke et al. (2020). Periodic signal analysis of such complex movements have shown to be important in real-world motion control tasks, contributing to the optimization of complex movements for robotic systems Shao et al. (2021); Feng et al. (2023). Building on the idea of periodic representations, Starke et al. (2022) developed a neural network structure that is capable of learning periodic characteristics from unstructured motion datasets. However, this model can only be trained on one type of motion data at a time. In addition, the phase information obtained is not well aligned with semantic signals, such as textual descriptions, limiting intuitive motion generation. Inspired by these observations, we propose PhaseFusion, a framework that incorporates periodic motion representations with diffusion models to achieve text-driven generation of a wide range of periodic natural movements of arbitrary lengths.

## 3 METHOD

Given input text descriptions as motion control indicators $c$, our goal is to generate natural and expressive motion sequences $J$ of arbitrary lengths with smooth transitions between adjacent synthesized motion sequences. To achieve this goal, our key idea is to transform the motion space into a learned phase manifold. The phase manifold is then used for motion synthesis and transition.

Same as DeepPhase (Starke et al., 2022), we assume that full-body movements inherently exhibit local spatial-temporal periodicity and use a phase representation to encode the input motions. However, directly applying the motion encoding method in DeepPhase to our setting fails to produce high-quality results, due to three reasons: (1) DeepPhase is designed to train on a specific type of motions and its performance degrades when training on diverse motions; (2) DeepPhase tends to encode low-frequency components of the motion, leading to limited accuracy in motion encoding; (3) The severe ambiguity in text-to-motion generation.

To address these issues, we present a compact yet expressive phase encoding with high-frequency information encoded. We then propose a new motion diffusion model for predicting phase parameters from input text descriptions to address the ambiguity issue. Compared to two widely-used motion representations, i.e., full skeletal poses and an one-dimensional code, phase encoding has a good balance in expressiveness and compactness. As illustrated in Figure 2, we first learn a network encoder to map the motion space to a periodic parameterized phase space. The encoder is trained by minimizing the distances between the original motions and the motions reconstructed from periodic parameters using the Inverse FFT (3.2). Next, we learn a conditional diffusion model to predict the periodic parameters, which is conditioned on the text prompt and the starting pose (3.3).

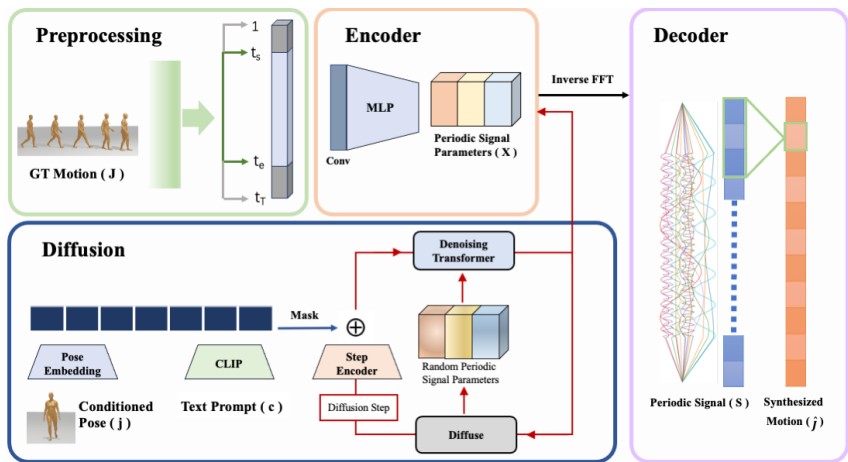

Figure 2: Overview of PhaseFusion. We first learn a network encoder to transform the motion space into a learned periodic parameterized phase space by minimizing the reconstruction errors between the original motions and the motions formed by decoding periodic parameters via Inverse FFT. Next, we train a conditional diffusion model to predict the periodic parameters with text prompts and a starting pose as inputs. During inference time, we apply diffusion to predict the periodic parameters and then decode the motion from the signal.

### 3.1 PREPROCESSING: ISOLATING THE PRIMARY MOVEMENT IN MOTION VIDEOS

Most sequences in the dataset include separate segments at the beginning and end, showing the initiation and conclusion of the movement. These segments do not reflect the essence of the intended action itself. We enhance our method by excluding these segments from being encoded into the periodic phase manifold, while using the linear signals for encoding them. Hence, we focus on encoding the primary motion into periodic phase manifolds, and then proceed to extend them during the testing phase.

To efficiently and robustly detect main segments of motion, we first process the whole motion sequence from a motion dataset into a latent space, where each frame $i$ of the sequence has a feature vector $d_i$ that extracts temporal and spatial features of the pose at the frame. We tested two choices of embedding the motions. Please refer to our Suppl. Materials for more detail. Then we are able to select the start pose $t_s$ and end pose $t_e$ through:

$$\underset{t_s, t_e}{\arg\min} \; \lambda_1 \cdot ||d_{t_s} - d_{t_e}||_2 - \lambda_2 \cdot \frac{t_e - t_s}{t_T} - \lambda_3 \cdot ||d_{t_e} - d_{t_T}||_2 \qquad (1)$$

where $t_T$ is the actual length of this motion clip; $\lambda_1$, $\lambda_2$ and $\lambda_3$ are the coefficients. The first term encourages the poses at the selected start and end time to be similar, aiding us in encoding the primary motion as a motion circle. The second term ensures that the primary motion is of adequate length, while the third term excludes the separate segments.

**Periodicity-Based Data Augmentation** To increase the data size with various start and end poses for training the main diffusion network that is conditioned on the starting pose (See 3.3 for more detail), we select the top $W$ sets of $t_s$ and $t_e$ for each motion video from the dataset. This selection pool enables us to train the network with the same text prompt and on the same training video but with a different starting pose.

### 3.2 A PERIODIC LATENT REPRESENTATION FOR MOTION SEQUENCES

Now, we aim to train an encoder network $\mathcal{E}$ that can encode the latent representation parameters from motion sequences and a convolution-based decoder network $\mathcal{D}$ that is trained to reconstruct the input sequences from the encoded parameters. Our latent representation $X = \{a_i, p_i, o_i\}_{i=1}^M$ is itself the parameters of $M$ discrete periodic phases. In particular, $\mathcal{E}$ encodes the input video to the amplitudes $a_i$, phase shifts $p_i$, and offsets $o_i$ parameters of a frequency decomposition of this periodic latent representation. Then with frequencies provided, $X$ can be transformed to its time-domain phase signal $S = \{s_i\}_{i=1}^M$ via inverse Fourier. Prior work (Starke et al., 2022) trained a

periodic latent representation by extracting frequencies from motions, which were predominantly in the low-frequency range, and it specialized only in one activity type such as walking, dancing, or dribbling. Since we aim to generate a large variety of motion sequences for many different activities conditioned on unconstrained text descriptions, we need to process a training dataset with many types of actions. Hence we provide a set of positive integer frequencies $F = \{f_i\}_{i=1}^M$ and let an encoder network predict the $a_i$, $p_i$, and $o_i$ for each of the phases. Then the phase signal can be calculated as:

$$s_i(k)_{periodic} = [a_i \cdot \sin(2\pi(f_i \cdot k + p_i)) + o_i, \ a_i \cdot \cos(2\pi(f_i \cdot k + p_i)) + o_i] \tag{2}$$

where $k$ is the value of time control. We set the lowest frequency value $min(F)$ to 1: this enables embedding the non-repeatable parts of the motion. Next, we set the highest frequency $max(F)$ to be a common multiple of all frequencies in $F$, so that the pattern of the signals can repeat smoothly forever as $k$ increases. This thus permits generating sequences of any desired length $T$ simply by setting $k = 1, 2, ..., T$. We can regard the encoded motion sequence as one motion circle, and can smoothly transition to a new extrapolated motion circle.

Our improved encoding design boasts two primary advantages: 1) it facilitates the synthesis of a broader range of motions that surpass the training motion distribution; and 2) it fosters a continuous motion embedding in the frequency domain, promoting arbitrary-length motion synthesis and seamless motion transitions.

**Encoding the Periodic Portion:** For each motion sequence $J$ and a pair of its $t_s$ and $t_e$, the encoder network $\mathcal{E}$ produces $X = \mathcal{E}(J^{t_s:t_e})$. $\mathcal{E}$ consists of an input convolution layer to reduce the dimension of $J^{t_s:t_e}$, followed by fully connected layers. To calculate $S^{t_s:t_e}$ for all the frames in $t \in [t_s, t_e]$, we align $k = 0$ to the $t_s$, and $k = 1$ to the $t_e$ and perform the calculation in 2.

**Encoding the Non-Periodic Portions:** We separately represent the non-repeatable parts, i.e. $t \in \{[1, t_s], [t_e, t_T]\}$ using linear signals:

$$s_i(k)_{linear} = [k \cdot (a_i \cdot \sin(2\pi p_i) + o_i), \ k \cdot (a_i \cdot \cos(2\pi p_i) + o_i)] \tag{3}$$

where we align $k = 0$ to where $t$ is 1 and $T$; and $k = 1$ to where $t$ equals $t_s$ and $t_e$.

Finally, to combine the periodic and non-periodic portions of signals into one complete $S$, we simply concatenate the signals in time in the correct order: $[S^{1:t_s}, S^{t_s:t_e}, S^{t_e:t_T}]$

**Auto-Encoder Training Objective:** We reconstruct $\hat{J}$ from $S$ by a 1-D convolution-based decoder network $\hat{J} = \mathcal{D}(S)$. The loss function for training this encoder-decoder setup for reconstruction is

$$L_{reconstruct} = ||\hat{J} - J||_2^2 + \lambda||FK(\hat{J}) - FK(J)||_2^2 \tag{4}$$

where $FK$ indicates the forwards kinematics for the joints, and $\lambda$ is a hyper-parameter for adjusting the loss.

### 3.3 Text-Conditioned Diffusion for Parameterized Motion Synthesis

After training the encoder $\mathcal{E}$ and the convolution decoder $\mathcal{D}$, we fix the weights of these two networks and use them as a foundation to train our diffusion module that produces $X$ from a text prompt $c$ and a conditional pose $j$. Providing our PhaseFusion model with a variety of conditional poses can improve comprehensive coverage of diverse character motions during training, and enhance its capability to generate videos from different starting pose. It also offers a clear and coherent starting point when concatenating motions. Facilitated by our periodicity-based data augmentation, we select a $t_s$ from the selection pool and use the pose at $t_s$ to help generate this periodic signal, and this conditional pose can be randomly masked during the training for performing prediction conditioned on $c$ only.

We denote $X^n$ as the periodic parameters at noising step $n$. For each $J$ and a pair of its $t_s$ and $t_e$ in the training set, we derive $X^0 = \mathcal{E}(J^{t_s:t_e})$. The diffusion noising step is a Markov process that follows:

$$q(X^n|X^{n-1}) = \mathcal{N}(X^{n-1}; \sqrt{\alpha_n}X^n, (1 - \alpha_n)I) , \tag{5}$$

where $\alpha$ is a small constant that could approach $X^n$ to a standard Gaussian distribution. According to Ho et al. (2020), this diffusion process can be formulated as:

$$q(X^n|X^0) = \sqrt{\bar{\alpha}_n}X^0 + \epsilon\sqrt{1 - \bar{\alpha}_n} , \tag{6}$$

where $\bar{\alpha}_n = \Pi_{i=0}^n \alpha_i$ and $\epsilon \sim \mathcal{N}(0, I)$. For the reverse step of this diffusion, we need to achieve denoising $\hat{X}^0$ from $X^n$, and we follow the idea in (Ramesh et al., 2022; Tevet et al., 2022) where we directly predict $\hat{X}^0$ using our denoising network $\mathcal{T}(j, c, n, X^n)$ conditioned on pose $j$, text prompt $c$ and noising step $n$ during training. To evaluate the quality of $\hat{X}^0$, we need to process it to achieve the time-domain signals $\hat{S}$, and then apply $\mathcal{D}(\hat{S})$ to reconstruct the motion sequences $\hat{J}$. We use the loss function to constrain $\hat{X}^0, X^0$ and $\hat{J}, J$ in training this part of the network.

### 3.4 GENERATING COMPOSITE AND EXTENDED MOTIONS WITH PHASEFUSION

Having trained our PhaseFusion model as above, we now discuss how to generate motion sequences. In the test stage, PhaseFusion is given $c$ and hyper-parameter $t_s, t_e$. For generating the first period of signals, we can start with $\emptyset$ or a random pose as the conditional pose, and we can use the end pose of the last period for smooth transitions between the periods. The sample stage of diffusion is similar to the approach seen in (Ho et al., 2020; Tevet et al., 2022). We start with $X^N \sim \mathcal{N}(0, I)$ iteratively execute the prediction $\hat{X}^0 = \mathcal{T}(j, c, n, X^N)$, and then this is diffused back to $\hat{X}^{n-1} \sim q(\hat{X}^{n-1}|\hat{X}^0)$ during the noising step $n$. The whole process iteratively decreases from $N$ to 1. Also, we apply the classifier-free guidance in (Ho & Salimans, 2022), that is to randomly mask out $c$ and $j$ during the training and then set a trade-off between $\mathcal{T}(j, c, n, X^N)$ and $\mathcal{T}(\emptyset, \emptyset, n, X^N)$ during sampling.

There are two operations for synthesizing new composited motions in our setting: (a) phase repetition; and (b) transition between two phases.

**Phase repetition** refers to extending the signals by allowing the time control value k to grow thus making the periodic part of $T$ repeat.

**Transition between two phases** refers to applying diffusion to synthesize a new set of parameters for extending motions. To ensure smooth transitions at conjunctions, the synthesis of control parameters is made to be conditioned on the initial pose, which can be set to the end human pose of the preceding period motion. This measure facilitates the alignment of the starting pose of the second phase with the ending position of the preceding phase. We also explore the ability of our framework to synthesize motions under different conditions to allow for concatenating actions with different semantic meanings.

## 4 EXPERIMENTS

In this section, we evaluate our PhaseFusion framework on the text-driven motion generation task and motion extension. In all the evaluated benchmarks, PhaseFusion achieves SoTA performances. In particular, when generating videos with longer time lengths, our method produces better results than previous methods. Moreover, the quality of our results would not decrease with time increasing.

### 4.1 EXPERIMENTAL SETTINGS

**Dataset** We evaluate our approach on the two most widely used datasets for text-driven motion generation tasks: KIT Motion Language (KIT-ML) (Plappert et al., 2016) and HumanML3D (Guo et al., 2022). KIT-ML includes 3,911 unique human motion sequences alongside 6,278 distinct text annotations. The motion sequences are at the frame rate of 12.5 FPS. HumanML3D is a more comprehensive dataset of 3D human motions and their corresponding text descriptions. It comprises 14,616 distinct human motion capture data alongside 44,970 corresponding textual descriptions. All the motion sequences in KIT-ML and HumanML3D are padded into 196 frames in length.

Following the settings used in previous works, the following five performance metrics we use are R-precision, Frechet Inception Distance, Multimodal distances, and Multimodality. For the details of these metrics. please refer to our Suppl. Materials.

**Implementation** The encoder network for generating periodic signals from motion is a 4-layered MLP with a convolution layer at the front. The decoder of the periodic signals is a 4-layered 1D convolutional network. Motions are encoded into 128 phases with the maximum frequency set to 30. We use a 512-dimension latent code to encode the conditioned pose and the text prompt, where the conditional pose is processed through an embedding layer, and the text feature is processed from a

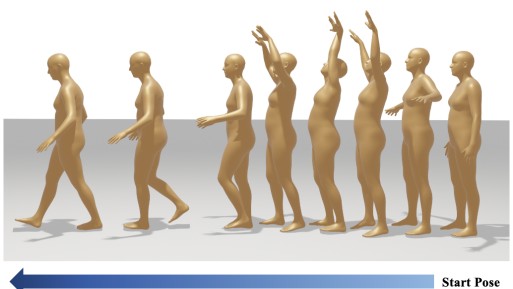 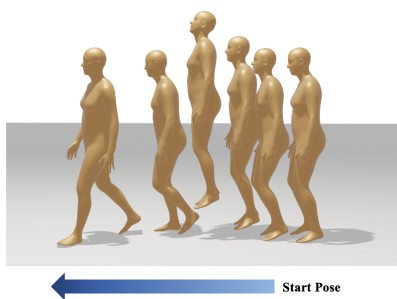

Figure 3: **Left:** The results of blending two motions generated from "A person raises their arms high above their heads." and "A person walks forward at medium speed." **Right:** The results of blending from "A person hops forward." and "the person is slowly walking ahead."

frozen CLIP-ViT-B/32 model. An 8-layered transformer for handling the denoising step. Our models were trained with batch size 128 with a learning rate that decayed from 1e-4 to 1e-6.

## 4.2 QUALITATIVE RESULTS

**Motion Blending**   As we have introduced in Section 3.4, our method is designed to produce smooth transitions among signals generated from different stages. We can also apply these techniques to generate motions from different conditions with different semantic meanings. With the conditioned pose and the decoder network $\mathcal{D}$ with multiple layers of convolutional operations, we are able to query two sets of phases that can be blended to produce smooth transition movements. We show some examples in Figure 3 and include more results in the supplemental materials.

**Interpolate the motions**   Because phase signals are continuous, we can actually consider the generation of longer motions using phase repetition as a process of extrapolating the phase. Similarly, we can also use phase to interpolate motions. By generating additional frames within the period of a generated motion, we can increase the frame rate of the motion video. Please refer to our Suppl. Materials for more detail.

## 4.3 COMPARISONS

For this comparison experiment, we select 7 related state-of-the-art methods: Language2Pose Ahuja & Morency (2019), Text2Gestures Bhattacharya et al. (2021), Dance2Music Aggarwal & Parikh (2021), MOCOGAN Tulyakov et al. (2018), Generating Diverse and Natural 3D Human Motions from Text Guo et al. (2022), MDM, Tevet et al. (2022), MotionDiffuse Zhang et al. (2022), T2M-GPT Zhang et al. (2023a) and MLD Chen et al. (2023). The quantitative comparison results on the HumanML3D test set and the KIT-ML test set are shown in Tables 1, respectively. Please refer to Suppl. Materials for visual results.

| Length | Method | Humanml | | | | | KIT | | | | |
|---|---|---|---|---|---|---|---|---|---|---|---|
| | | Top-3 R-Precision ↑ | FID ↓ | MM-DIST ↓ | Diversity → | MModality ↑ | Top-3 R-Precision ↑ | FID ↓ | MM-DIST ↓ | Diversity → | MModality ↑ |
| | Real | 0.797 | 0.002 | 2.974 | 9.503 | - | 0.779 | 0.031 | 2.788 | 11.08 | - |
| Standard Length | Language2Pose | 0.486 | 11.02 | 5.296 | 7.676 | - | 0.483 | 6.545 | 5.147 | 9.073 | - |
| | Text2Gesture | 0.345 | 5.012 | 6.03 | 6.409 | - | 0.338 | 12.12 | 6.964 | 9.334 | - |
| | Dance2Music | 0.097 | 66.98 | 8.116 | 0.725 | 0.043 | 0.086 | 115.4 | 10.40 | 0.241 | 0.062 |
| | MOCOGAN | 0.106 | 94.41 | 9.643 | 0.462 | 0.019 | 0.063 | 82.69 | 10.47 | 3.091 | 0.250 |
| | Guo et al. | 0.736 | 1.087 | 3.347 | 9.175 | 2.219 | 0.681 | 3.022 | 3.488 | 10.720 | 2.052 |
| | MDM | 0.611 | 0.544 | 5.566 | 9.559 | 2.799 | 0.396 | 0.497 | 9.191 | 10.847 | 1.907 |
| | MotionDiffuse | 0.782 | 0.630 | 3.113 | 9.410 | 1.553 | 0.739 | 1.954 | 2.958 | 11.100 | 0.730 |
| | T2M-GPT | 0.775 | 0.141 | 3.121 | 9.722 | 1.831 | 0.745 | 0.514 | 3.007 | 10.921 | 1.570 |
| | MLD | 0.772 | 0.473 | 3.196 | 9.724 | 2.413 | 0.734 | 0.404 | 3.204 | 10.800 | 2.192 |
| | Ours | 0.761 | 0.397 | 3.194 | 9.363 | 2.805 | 0.736 | 0.503 | 3.167 | 10.873 | 2.071 |
| 3x Standard Length | Guo et al. | 0.496 | 8.701 | 4.947 | 7.370 | - | 0.551 | 7.591 | 4.907 | 8.767 | - |
| | MDM | 0.121 | 18.499 | 9.706 | 5.806 | - | 0.094 | 41.760 | 11.228 | 5.588 | - |
| | MotionDiffuse | 0.392 | 10.303 | 5.819 | 5.928 | - | 0.281 | 72.225 | 9.961 | 6.119 | - |
| | MLD | 0.546 | 7.842 | 4.976 | 7.938 | - | 0.495 | 7.681 | 5.335 | 9.235 | - |
| | Ours (Repetition) | 0.751 | 0.385 | 3.266 | 9.009 | 2.802 | 0.727 | 0.902 | 3.429 | 10.067 | 1.968 |
| | Ours (Generative) | 0.725 | 0.584 | 3.311 | 8.884 | 2.817 | 0.710 | 1.174 | 3.483 | 9.813 | 2.196 |

Because in the 3× *Default Length* experiments, the baseline methods easily result in random motion failure, the MModality results become unreliable. Therefore, we only report the MModality of our method when generating 3×default length.

Table 1: Results for HumanML3D and KIT test set. The best, second-best, and third-best results are highlighted in red, orange, and yellow, respectively. Our method achieves comparable results to the baseline methods in the Default Length setting and significantly better results in the 3× Default Length.

Our method is evaluated in two settings. **Default Length**: We consider the 196-frame motion length used in previous works as the default length setting. $3\times$ **Default Length**: To further evaluate the quality of generated motion for extended lengths, we generate motions of $3\times$ default length and select the last 196 frames for evaluation. In particular, we test two different operation modes for generating longer sequences using our method. For the *Repetition* mode, we apply our diffusion model once to produce one set of parameters that determine a repetition period and repeat the sequence to form a sequence with $3\times$default length frames. For the *Generative* mode, we call our diffusion model multiple times until the length of the (accumulated) generated sequence reaches $3\times$default length. We compare our approach with the top performing methods in the $3\times$ *Default Length* setting and can manually set the length of generated motion, i.e., Guo et al. (2022), MDM Tevet et al. (2022), MotionDiffuse Zhang et al. (2022), and MLD Chen et al. (2023).

From the tables 1 , we can see that our method demonstrates comparable performance in the *Default Length* setting and a significant advantage over the baselines in the $3\times$ *Default Length* setting.

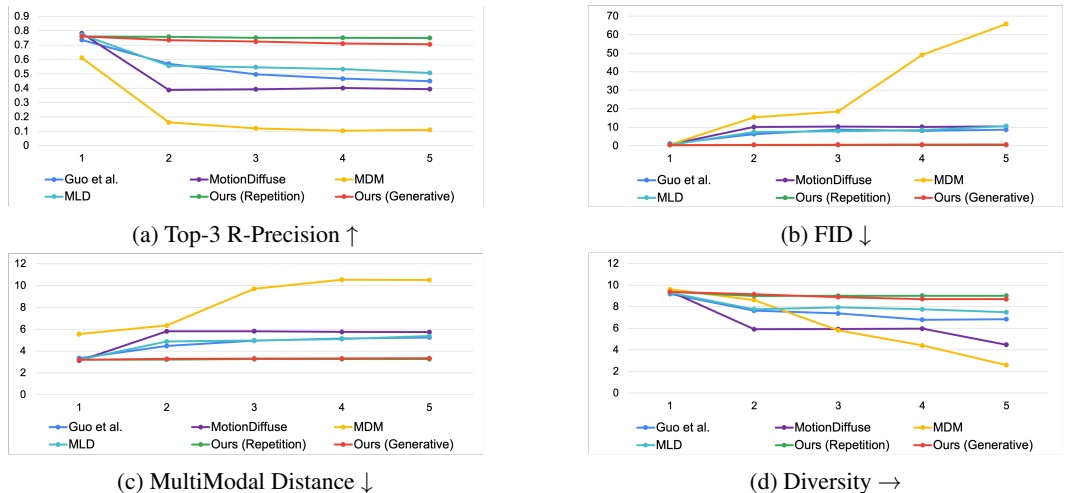

(a) Top-3 R-Precision $\uparrow$        (b) FID $\downarrow$

(c) MultiModal Distance $\downarrow$        (d) Diversity $\rightarrow$

Figure 4: Effect of motion sequence length on the generation quality on HumanML3D test set. Our method is much more stable than other methods for long sequence generation. The horizontal axis represents the length setting, and the horizontal axis represents the metric.

Furthermore, to better examine the effect of the generated sequence length on the generation quality, we conduct the same experiments on the HumanML3D test set for length settings from 1 to 5 times the default length setting and plot the Top-3 R-Precision, FID, MultiModal Distance, and Diversity scores as a function of motion length. As shown in Figure 4, compared to other methods that exhibit immediate performance deterioration once the length setting increases, the quality of the sequence generated by our method is nearly length-invariant. This indicates that our method can generate sequences of arbitrary lengths with stable quality.

Here we analyze why the baseline methods have difficulties in generating long sequences. For those methods that use diffusion directly, i.e., Tevet et al. (2022); Zhang et al. (2022), long lengths are outside of the training distribution, and the quality is impacted immediately since the denoising module must handle the whole sequences at once. Recurrent network-based methods, i.e., Guo et al. (2022), have the ability to handle variable lengths using an auto-regressive approach, but typically fail for other reasons; for example, the quality of generated outputs deteriorates slowly over time. Our method generalizes much better to longer generation lengths and unseen motions that are far from the training motion distribution.

## 4.4 ABLATION STUDIES

**Motion Encoding**    We conduct an ablation study on the parameterized phase representation used in PhaseFusion. As shown in Table 2, we evaluated different combinations of the choices of phase signals concerning the highest frequency, the representation of the phases and the number of phases being used, and measure the quality of the reconstructed motions using our encoder $\mathcal{E}$ and decoder $\mathcal{D}$.

The result shows that using both sin and cos functions and adding high frequency phases can help encode the high-frequency details of motions.

| Method | | | Metrics | | |
|---|---|---|---|---|---|
| Highest Freq | Representation | Num of Phases | Top-1 | Top-3 | FID |
| 8 | sin | 128 | 0.457 | 0.784 | 0.232 |
| 30 | sin | 128 | 0.499 | 0.789 | 0.106 |
| 30 | sin, cos | 128 | 0.506 | 0.791 | 0.082 |
| 30 | sin, cos | 256 | 0.507 | 0.792 | 0.080 |
| DeepPhase | | | 0.426 | 0.737 | 0.968 |

Table 2: Reconstruction results of using different settings of phases. The best and second-best results are highlighted in red and orange, respectively. We set our configuration to the highest frequency of 30, using both sine and cosine representations, with a total of 256 phases, because improvement gained from setting the number of phases to 256 is marginal, and using more phases may also lead to a decline in computational speed.

**Pose Condition-based Phase Transitions** We also present an ablation study on the different pose conditions choices for translating between different phases, i.e., the choice for $j$ when using $\mathcal{T}(j, c, n, X^n)$ for generating new phases.

We choose a test batch contains 32 different text prompts from HumanML3D Guo et al. (2022). For each text prompt, we first use it to produce the starting motion sequence, then we randomly choose another text prompt to generate the ending motion sequence. The generation of the ending sequences is tested under three different pose conditions: 1) the last poses of the starting sequences; 2) random poses; 3) zero masks.

To evaluate the quality of the transitions between these two motion segments, we use the encoder network $\mathcal{E}$ and the decoder network $\mathcal{D}$ to reconstruct the new motion with transition frames. The networks should reconstruct a motion sequence with better quality if the two segments of the sequence are smoothly transitioned because such a motion sequence is closer to the training samples. Therefore, we

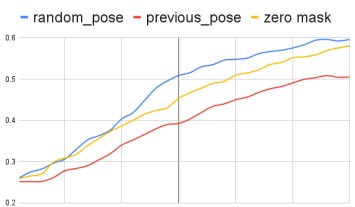

Figure 5: Per-frame distances between the generated motions and the reconstructed motions by the autoencoder. The horizontal axis indicates the frame index: 0 is the moment when the phase signals of the first sequence switch to those of the next sequence. The vertical axis indicates the pose distances.

measure the L2 distance between the input and reconstructed motions among the frames centered at the transition moment. And this distance can serve as a metric to evaluate the quality of the transitions. From 5, we can see that if the phase parameters of the ending sequences are generated conditioned on the last pose of the starting segments, then the synthesized motion would have better qualities.

## 5 CONCLUSIONS

In this paper, we proposed a new method for generating high-quality human motions in arbitrary lengths for text descriptions. We propose to encode repetitive motions using periodic signals and design a new diffusion model to predict the periodic signals given text inputs. Our method achieves natural motion generation with smooth transitions, especially for long-sequence generation, and significantly outperforms previous methods.

**Limitation and future work** At present, our method cannot directly extract and understand the quantitative vocabularies (e.g., "walk for 11 steps" and "turn left 30 degrees") provided in text prompts but only treat them equivalently to regular descriptions, resulting in an inability to generate precise motions for these text commands. Therefore, a meaningful future direction would be to extract specific quantitative control commands from text, and directly use these commands to operate periodic signals and generate more controllable motions. It is also interesting to explore how to incorporate the environmental constraints (e.g., uneven terrain and large obstacles which require the character to respond accordingly) into our motion generation framework to enable its practical use in gaming and virtual simulation engines.

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

SUPPLEMENTARY

# 1 REPETITIVE PERIOD DETECTION

In section 3.1, we introduce a method of locating the starting time $t_s$ and the ending time $t_e$ of the longest primary motion for each ground truth motion sequence. Specifically, we train an Auto-Encoder to encode the whole videos into a sequence of features, where for the pose of each frame $i$ there is a corresponding feature vector $d_i$ that encodes the temporal and spatial information. Then $t_s$ and $t_e$ can be determined via 1. We evaluate two kinds of phase embedding when training the Auto-Encoder: the original DeepPhase and our novel configuration detailed in 3.2. 1 and 2 present Euclidean loss matrices where both the horizontal and vertical axes represent time frames; the value at matrix(i, j) indicates the difference between frame i and frame j. Both embedding methods effectively highlight the primary movement segments. Nonetheless, with intentionally incorporated high-frequency components, Figure 2 exhibits clearer boundaries for areas of interest, facilitating the pinpointing of distinct $t_s$ and $t_e$ pairs. Here we show some cases of selected $t_s$ and $t_e$ in 1 and 2.

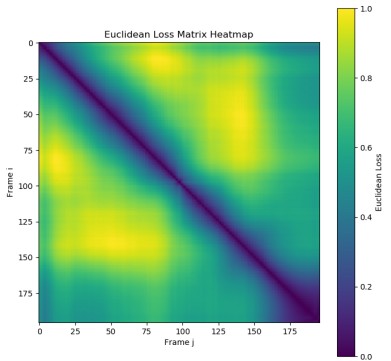 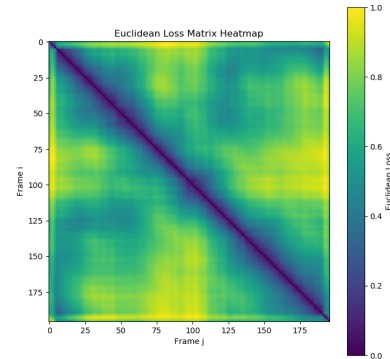

Figure 1: The Euclidean loss matrices of features extracted using DeepPhase.

Figure 2: The Euclidean loss matrices of features extracted using our novel design.

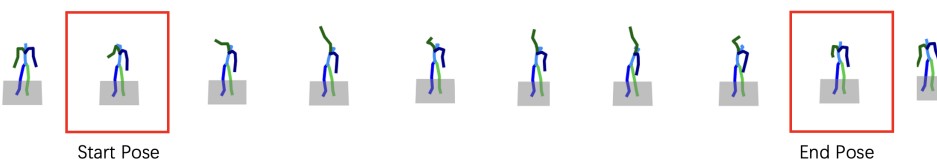

Figure 3: Example ground truth motion sequence with the text description of "the person is stretching his right arm.". We show the starting pose and the ending pose at $t_s$ and $t_e$.

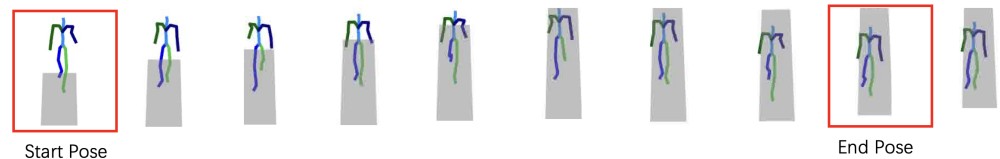

Figure 4: Example ground truth motion sequence with the text description of "a person walks straight forward at moderate speed". We show the starting pose and the ending pose at $t_s$ and $t_e$.

## 2 INTERPOLATE THE MOTIONS

To increase the frame rate of a motion video, we can utilize phase interpolation. When creating videos at the original frame rate, we extract $t_T$ samples from the time-domain signal $S = [S^{t_1:t_s}, S^{t_s:t_e}, S^{t_e:t_T}]$. By multiplying the sample rate by a factor of $N$, we can boost the video's FPS. This process involves distributing the samples obtained from the high sample rate across $N$ distinct signal sets, each representing a full motion sequence in the original frame rate. Then we apply our decoder $\mathcal{D}$ to each of these signal sets and subsequently rearrange the resulting frames in alignment with their initial sampling order. Through this method, we effectively interpolate a motion video from $t_T$ frames to $N \times t_T$ frames. Please refer to our supplementary video for the visual result.

## 3 EVALUATION METRICS

Following the settings used in previous works Guo et al. (2022), we use the following five performance metrics: ***R-Precision*** evaluates a motion sequence alongside 32 textual narratives, including 1 accurate one and 31 other arbitrary descriptions. This involves calculating the Euclidean distances between the embeddings between the motion and the textual content. We then report the accuracy of the motion-to-text retrieval at the top-3 level. ***Frechet Inception Distance (FID)*** is also used to evaluate the quality of motion generated by generative methods. It measures the distance between the real and generated distributions in the feature space of a pre-trained model. ***Multimodal distances (MM-DIST)*** quantify the distances between the text derived from the provided description and the generated motions. ***Diversity*** is measured by arbitrarily dividing all the created sequences from all testing texts into pairs, and the average combined differences within each pair are determined. ***Multimodality (MMmodality)*** is measured by generating 32 motion sequences in a single text description and calculating the differences within these consistently produced sequences. Same as previous works, we evaluate all the metrics by using the pre-trained network from Guo et al. (2022) for gathering the text and motion embeddings.

## 4 EFFICIENCY ANALYSIS

Generating longer sequences leads the computation cost of attention operations to increase quadratically for methods using latent dependent to motion length since all pairwise interactions are calculated. While using compact frequency-domain parameters $X$ as our latent variables, PhaseFusion can efficiently produce high-quality motion of any desired length. We test the average inference time for one sentence using an A6000 GPU with different motion length settings. The results are shown in 5.

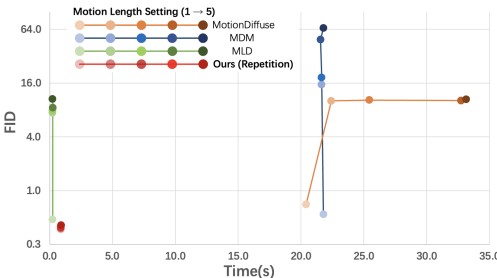

Figure 5: Average inference time for processing one sentence with different motion length settings. Note that only MLD Chen et al. (2023) and ours are using fixed length latent, and the shape of the latent are $1 \times 256$ and $3 \times 128$ respectively.

## 5 CLASSIFIER-FREE GUIDANCE PARAMETERS

In this work, we follow Tevet et al. (2022), Chen et al. (2023) and Ho & Salimans (2022) that use a hyperparameter $s$ to balance between $\mathcal{T}(j, c, n, X^N)$ and $\mathcal{T}(j, \emptyset, n, X^N)$ during sampling, where the adjusted $\mathcal{T}'(j, c, n, X^N)$ is given by:

$$\mathcal{T}'(j, c, n, X^N) = \mathcal{T}(j, \emptyset, n, X^N) + s \cdot (\mathcal{T}(j, c, n, X^N) - \mathcal{T}(\emptyset, \emptyset, n, X^N)) \qquad (1)$$

Then we scale $s$ from 1.5 to 5.5 and evaluate the sampling results in the test set of HumanML3D. The results in 6 indicate that setting h to the range between 2.5 to 3.5 would yield the most balanced performance across the indicators.

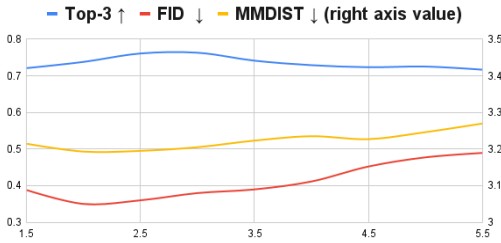

Figure 6: Results of R Precision (Top-3), FID and Multi-modal Distances (MMDIST) for different $s$. The horizontal axis indicates the value of $s$. The vertical axis on the left indicates the values for R Precision and FID, while the vertical axis on the right indicates the values for MMDIST.

## 6    ADDITIONAL VISUAL RESULTS

Please refer to our supplemental video for more visual results. In the presented video, we illustrate a comparison between standard length generation and extended length generation; the transitions between various phases under both the same and different textual conditions; motion interpolation; and an instance of long motion generation.

