# OpenReview forum: "PhaseFusion: A Diffusion-based Periodic Parameterized Motion Generation Framework"
_ICLR.cc/2024/Conference — ICLR 2024 Conference Withdrawn Submission_

### Official Review · Reviewer_DwHH · 2023-10-30

**Soundness:** 3 good
**Presentation:** 3 good
**Contribution:** 2 fair
**Rating:** 6
**Confidence:** 2

**Summary:**

The paper present an approach to diffusion models utilizing periodic random signal to better generate human motion, in particular for extended length sequences. This approach is empirically shown to improve upon existing state of the art as the length of generated sequence increases.

**Strengths:**

The authors do a good job motivating and presenting the proposed approach. The proposed approach not only introduces a novel periodic encoder but additionally a strategy to augmenting data to provide increased diversity during training.

Empirically the proposed approach appears to be strong, in particular for extended length sequences.

**Weaknesses:**

The two main areas of concern are the empirical evaluations and potentially narrow scope.

The empirical results are interesting, however there are concerns related to the setup and evaluation. The data used to train the proposed approach is unclear, with only the evaluation data sets mentioned. For the extended length sequence, evaluation is only performed on the last 196 frames, failing to capture the performance difference in the earlier frames. Additionally reporting an average over strided evaluations would further demonstrate the improvement of the model. Additionally it would be interesting to see the impact of excluding the data augmentation strategy to demonstrate the performance improvement due to the proposed model versus this data augmentation.

With respect to the scope, it appears the main value of the proposed approach is an improvement in performance for extended length sequences for motion generation, as at the standard length the proposed approach does not appear to significantly outperform existing models. This not only limits the potential audience of interest but additionally would be of more interest if corresponding annotated datasets with longer sequence lengths existed in order to better evaluate rather than cropping longer generated sequences to evaluate utilizing shorter annotated data.

**Questions:**

Can more details on the data used to train the proposed model be provided?

---

### Official Review · Reviewer_jvFF · 2023-10-31

**Soundness:** 3 good
**Presentation:** 2 fair
**Contribution:** 3 good
**Rating:** 3
**Confidence:** 4

**Summary:**

This work is for the task of text-guided motion generation. The core idea is to encode motion into a phase manifold, which is a more compact representation space. This is reasonable since motion itself has repetitive components. At test-time, they use a conditional diffusion model to generate the Periodic signal parameter. In this way, they can generate motion according to the given text.

**Strengths:**

1. Within the scope of diffusion-based human motion generation research, this is the pioneering work that encodes motion into the phase manifold space.

2. This approach is capable of producing long motion sequences under text guidance.

**Weaknesses:**

1. It's challenging to assess the actual impact of this study. The authors assert their ability to generate diverse and arbitrary-length motion sequences. However, only two samples are displayed in Fig. 3, both of which appear relatively simple. While the supplementary material offers more results, comparisons with other works are absent.

2. The paper lacks a detailed explanation of how the generation of long and complex motion is achieved, especially regarding motion transition. The diversity also seems somewhat limited.

3. There are several issues with the writing:
- Instead of "using the Inverse FFT (3.2)", it should be "using the Inverse FFT (Section 3.2)". Similarly, "the starting pose (3.3)" should be rephrased as "the starting pose (Section 3.3)".
- The images in the figures are somewhat blurry, and the text font size is too small for comfortable reading.
- In Table 1, shouldn't the results under "Real" in "Humanml" be centered?
- The paper does not provide illustrations or explanations on how the evaluation metrics were calculated.

I believe the paper's idea is promising. However, this version seems hastily put together. From the writing to the experiments, the authors should give them a thorough review and polish.

**Questions:**

[1] Can this work generate a more complex sequence, such as walking --> clamp -->  sit?

[2] Since Diffusion is a probability model, could you show a few more samples under the same text guidance?

[3] How to process the transition of two different motion categories?

[4] Can you give some visualization of the learned phase parameters? For example, only decode the first 10 components.

---

### Official Review · Reviewer_N4P9 · 2023-10-31

**Soundness:** 2 fair
**Presentation:** 2 fair
**Contribution:** 3 good
**Rating:** 3
**Confidence:** 3

**Summary:**

PhaseFusion presents a Diffusion-based text-to-human-motion method. It consists of a learned Decoder which takes as input a periodic signal and produces human motion and a Diffusion model which produces the periodic signals for motion generation. The method produces strong results on two datasets and the authors show that the method is able to generate motion for much longer time horizons that other SOTA methods.

**Strengths:**

Learning human motion in phase space for motion synthesis is novel and clever. The method produces strong results on standard benchmarks and the authors verify the effectiveness of the approach for long time horizon motion generation.

**Weaknesses:**

I have two major concerns: (A) the method is not well-presented and (B) the authors make many claims but do not substantiate them in text nor in ablations.

(A) Overall, the method is not well-presented:
* the authors extend on DeepPhase and the paper requires an understanding of this work - the authors should have briefly described DeepPhase in equations at the beginning of Section 3 - this would have been a great opportunity to also differentiate their work from DeepPhase
* A method overview is missing and Figure 2 is insufficient for this. For example, it only becomes clear in Section 3.3. on page 5 that the authors utilize a multiple training stages.
* Figure 2 is unclear: what are the different steps and how are they connected? This could be improved by adding better descriptions in text and in the figure, i.e. linking the Figure with the Equations and text.
* In Section 3.3 diffusion is briefly introduced. However, only the forward step is presented in equations while the backward step- arguably the more relevant part for this work - is not.
* The last Sentence of 3.3. is unclear: “what” loss function is used to train “what” part of the network?

The method has many steps, starting at the pose preprocessing, pre-training of the encoder decoder model and finally the diffusion model. I strongly suggest that the authors first start Section 3 with an overview of the method, outlining the general idea, explaining that the encoder-decoder are retrained and how the pre-processing works. The method is not well-descried in Equations but rather in text. This makes it really difficult to follow. I strongly suggest to rely more on Equations - those can be used not just in text but also for example in Figure 2, linking the overview image with the text.

The authors claim that they utilize a “compact yet expressive … phase space” (Abstract) but in Section 3.2 it seems that latent “phase” representation X has the same amount of phases as pose (?) signal S has time steps - so it seems that no high-frequency signals are actually removed and no compression is happening.

(B) The paper makes many claims which are not substantiated in text or ablation:
* Page 2: “By encoding motion in … periodic phase space we alleviate the need for large amounts of training data” - How? This is not confirmed by any experiments and I don’t see how that would help with missing data. This claim should be explained in more detail and also ablated.
* Page 3: “(2) DeepPhase tends to encode low-frequency components…. leading to limited accuracy” - why?
*  Page 5: “it facilitates the synthesis of a broader range of motions that surpass the training motion distribution;” how is this verified? This is a very big claim
* Page 5: “it fosters a continuous motion embedding in the frequency domain” - could the authors expand on what the mean here by “continuous motion embedding”?


Minor:
* Figure 4 is missing x- and y-axis labels.
* The authors sometimes use the term “video” (e.g. Section 3.2) when I believe they mean “human motion sequence”?
* In 3.1 replace “... feature vector $d_i$ that <extracts> temporal …” with “... feature vector $d_i$ that <contains> temporal …”

**Questions:**

What is the limit of the generation length? For example, in Figure 1, can the circle be closed and looped over if the number of frames increases? Would the circle stay the same size?

I find the last video sample very impressive: is this one continuous generation or are the prompts fed into the model separately and the start- and end-poses are fit between the different generations?

---

### Official Review · Reviewer_psKt · 2023-11-01

**Soundness:** 2 fair
**Presentation:** 2 fair
**Contribution:** 3 good
**Rating:** 5
**Confidence:** 4

**Summary:**

This paper introduces a diffusion-based text-conditional human motion generator in the frequency domain. The motion is encoded into a periodic parameterized space similar to DeepPhase, whereas the task of this paper is challenged with high-frequency details that do not originate from DeepPhase. The diffusion model aims to synthesize the phase parameters based on textual descriptions and onset poses. The results show that the method proposed in this paper can generate long-term, smooth motion.

**Strengths:**

1. as stated in the paper, their proposed method is the first to use phase in a text-conditional motion generation task.

2. the original DeepPhase paper focused on generating motion with only low-frequency detail and small datasets. This paper proposes several techniques to address this gap.

3. experimental results show the merit of the methods proposed in this paper.

**Weaknesses:**

1. The paper claims that existing methods have difficulties in generating motion sequences of arbitrary length with respect to motion diversity and smooth transitions, and I agree that phase diversity might be easy to lead to more diverse motion sequences. However, for smooth transitions generated over long periods of time, there are existing methods that combine interpolation with diffusion modeling that achieve good performance, not limited to [1,2,3,4].

2. in line with the above, I assume that the baselines used for long-term generation (e.g., MDM) take advantage of the fact that transformer architectures can accept different input lengths (and if possible, please confirm this in the revision as well), whereas I do believe that the methods mentioned above, autoregressive generation and interpolation during diffusion is feasible, as this is already a very popular tool [1,2,3,4], which should be compared to as well.

3. The proposed method seems to require an initial pose as input and baselines do not. If this difference is taken into account when performing the quantitative evaluation, then the comparison would not be fair.

4. In the demo video provided, I do not see complex motion in HumanML3D, such as backflips or crawling on the ground, I suspect that the proposed phase-based approach is still ineffective for such less periodic movements with high-frequency details.

[1] Raab et al. Single Motion Diffusion. Arxiv 2023

[2] Shafir et al. Human motion diffusion as a generative prior. Arxiv 2023

[3] Zhang et al. DiffCollage: Parallel Generation of Large Content with Diffusion Models. CVPR 2023

[4] Tseng et al. EDGE: Editable Dance Generation From Music. CVPR 2023

**Questions:**

1. "phase encoding has a good balance in expressiveness and compactness", is there any evidence showing that phase encoding is better in expressiveness than full skeleton pose or SMPL parameters?
2. "it facilitates the synthesis of a broader range of motions that surpass the training motion distribution", is there any reason?

Overall, the author's response to the concerns in the Weakness section is needed to make the final decision. I am happy to increase the rating if my concerns are addressed.